# Sites of high local frustration in DNA origami

Richard Kosinski[1], Ann Mukhortava[2], Wolfgang Pfeifer [1], Andrea Candelli[2], Philipp Rauch[2] & Barbara Saccà[1]

The self-assembly of a DNA origami structure, although mostly feasible, represents indeed a rather complex folding problem. Entropy-driven folding and nucleation seeds formation may provide possible solutions; however, until now, a unified view of the energetic factors in play is missing. Here, by analyzing the self-assembly of origami domains with identical structure but different nucleobase composition, in function of variable design and experimental parameters, we identify the role played by sequence-dependent forces at the edges of the structure, where topological constraint is higher. Our data show that the degree of mechanical stress experienced by these regions during initial folding reshapes the energy landscape profile, defining the ratio between two possible global conformations. We thus propose a dynamic model of DNA origami assembly that relies on the capability of the system to escape high structural frustration at nucleation sites, eventually resulting in the emergence of a more favorable but previously hidden state.

[1] ZMB, University of Duisburg-Essen, Universitätstr. 2, 45117 Essen, Germany. [2] LUMICKS, De Boelelaan 1085, 1081 HV Amsterdam, The Netherlands. Correspondence and requests for materials should be addressed to B.S. (email: barbara.sacca@uni-due.de)

The folding of a protein into a unique structure is driven by the information encoded in its sequence. In favorable conditions, protein folding occurs spontaneously and in a relatively short time; however, the target shape—evolutionary tailored to fulfill a defined function[1,2]—must be selected from an exponentially large conformational space and clearly cannot be the result of a random search. A possible solution to this paradox is captured by the so-called principle of "minimal frustration"[3] according to which the folding pathway of a protein towards its most probable configuration is biased by a rapid gain in energy stabilization for conformations progressively similar to the native state. Successive rounds of natural selection smoothen the roughness of the energy landscape enabling to overcome local asperities (kinetic traps) and pay for the loss in entropy of the polymer chain.

When applied to nucleic acids, free energy landscape formalisms have provided the conceptual frameworks to describe the thermally or mechanically induced transitions of Holliday junctions[4,5], DNA hairpins[6–8], and G-quadruplexes[9]. However, contrary to proteins, self-assembly of nucleic acids is facilitated by the predictable recognition of only four (rather than twenty) interaction partners. This drastically reduces the complexity of the folding problem and allows to establish a robust sequence-to-structure relationship, which is the essence of nucleic acids design and the workhorse of emerging DNA nanotechnology methods[10]. Among these, the DNA origami approach stands out for the extraordinary spatial control that enables the realization of molecular objects with nanosized features and structural complexity approaching those of natural protein constructs[11]. A DNA origami structure is the result of thermal annealing between a long single-stranded DNA chain (scaffold) and few hundreds of short DNA sequences (staple strands) that are designed to hybridize distinct and mostly discontinuous regions of the scaffold[12,13]. The final object has typically a mass of about 5 MDa and comprises more than 7000 base pairs. It is therefore clear that, despite the robust programmability of DNA, an underlying mechanism must exist that ensures the formation of only one species in a relatively short time and with surprisingly low error rates.

Theoretical and experimental studies along this line[14–26] allowed to reveal three main aspects of the problem: (i) distinct folding trajectories do exist, meaning that absolute and local minima of energy are present in the energy landscape of the reaction; (ii) the fate of the assembly may be affected by mechanical triggers that typically target the exposed regions of the structure, where the scaffold inverts its direction, i.e. the edges; (iii) upon addition of triggers, the mechanical transformations observed can be ascribed to the allosterically-driven reconfiguration of joined Holliday junctions (HJs), the fundamental units of every DNA origami structure. In all these studies, however, DNA strands are approximated to elastic polymers and the folding process is analyzed either in terms of statistical paths of worm-like chains or by mechanical coupling of connected motifs, omitting the crucial sequence-to-structure relation which is typical of natural self-assembling systems.

The aim of this work is to understand the role of sequence-coded information in the folding of monolayer DNA origami structures and how its combination with topology-dependent features may provide a unified view of the energetic contributions in play. For this purpose, we analyzed the assembly products of origami domains having identical structure but different nucleobase composition and compared the results obtained upon application of distinct extents of topological strain at the edges of the structure, where structural frustration is higher. Our data show that the folding pathway of a monolayer DNA origami structure has two minima of energy, corresponding to the two

possible right-handed isomers of all constituent crossovers (iso I and iso II in Fig. 1a). In our model, the ratio between these two states is dictated early in the assembly process and depends on the degree of mechanical stress experienced by the nucleation strands at the points of scaffold turn. In this sense, the self-assembly of a DNA origami structure is an adaptive process, i.e., it may change its fate by escaping the high structural frustration imposed by unfavorable scaffold routing at the initiation sites, leading to a reshape of the energy landscape and accumulation of a more stable isomer.

## Results

**Design of three quasi-independent DNA origami domains.** Canonical two-dimensional (2D) origami designs that follow the original Rothemund´s prototype[12] rely on the regular spacing of staples and scaffold crossovers to maximize the number of connected helices and minimize the strain at the scaffold turns (Suppl. Fig. 1). Deviations from these fundamental rules lead to energetic barriers that must be overcome to permit folding and have been recently employed to trigger the partial or complete reconfiguration of DNA origami structures from a canonical (iso I) state to a less common iso II isomer[15,17]. Both states share exactly the same sequences and are therefore composed of the same pairs of hybridized segments; however, the strands that meet at the crossover are different. Whereas in the iso I, antiparallel stretches of scaffold strands are kept in place by staple crossovers (Fig. 1a left), the iso II form relies on antiparallel stretches of staples joined by scaffold crossovers (Fig. 1a right). When extended to a pattern of bound HJs, this reflects into an interesting structural duality between the core and the edges of the two isomers. The cores are indeed related by an exchange in base stacking partners, with I/IV and III/II stacked pairs of every HJ in the iso I form becoming II/I and IV/III pairs in iso II (highlighted square in Fig. 1a–c). The geometry of the edges is also swapped, with left/right scaffold loops and bottom/up linear scaffold segments of iso I becoming, respectively, left/right linear segments and bottom/up loops in iso II (gray strands in Fig. 1b, c).

What governs the formation of one isomer over the other during origami folding? To address this question, we designed a monolayer DNA origami composed of three quasi-independent domains (named as A, B, and C) of identical structure but distinct sequence content (Fig. 2a) and analyzed the outcome of the assembly process in absence and presence of variable edge topologies. As during folding the three domains experience the same experimental conditions, any possible difference observed in their assembly fate—when subject to equal mechanical strains—can be uniquely associated to a sequence-dependent factor.

In our design, the core of each domain (e(−) construct) is constituted by 56 staples of equal length, organized into 45 antiparallel Holliday junctions flanked by unpaired scaffold stretches along the contour (Fig. 1b, first panel; Suppl. Figs. Figures 2 and 3). To target the edges of each domain, we designed two sets of staples, one for the left- and one for the right-side of iso I (yellow strands in Fig. 1b) such to induce scaffold inversions, respectively, at unfavorable and favorable positions (Suppl. Fig. 1). Each set of edges was designed in three distinct topologies (Fig. 1b; e(0), e(1) and e(2), second to the fourth panel, respectively). All share identical base stacking interactions with the adjacent core strands (Fig. 1b, blue and cyan strands, for the left- and right-side edges, respectively); however, their binding to the scaffold is different. Edges type 0 are constituted by two antiparallel helical stretches of equal length, forming a U-shape with the scaffold (Suppl. Fig. 4). Edges type 1 and 2 instead hybridize to three different segments of the scaffold through an additional T5 hinge, resulting in the formation of an S- and an

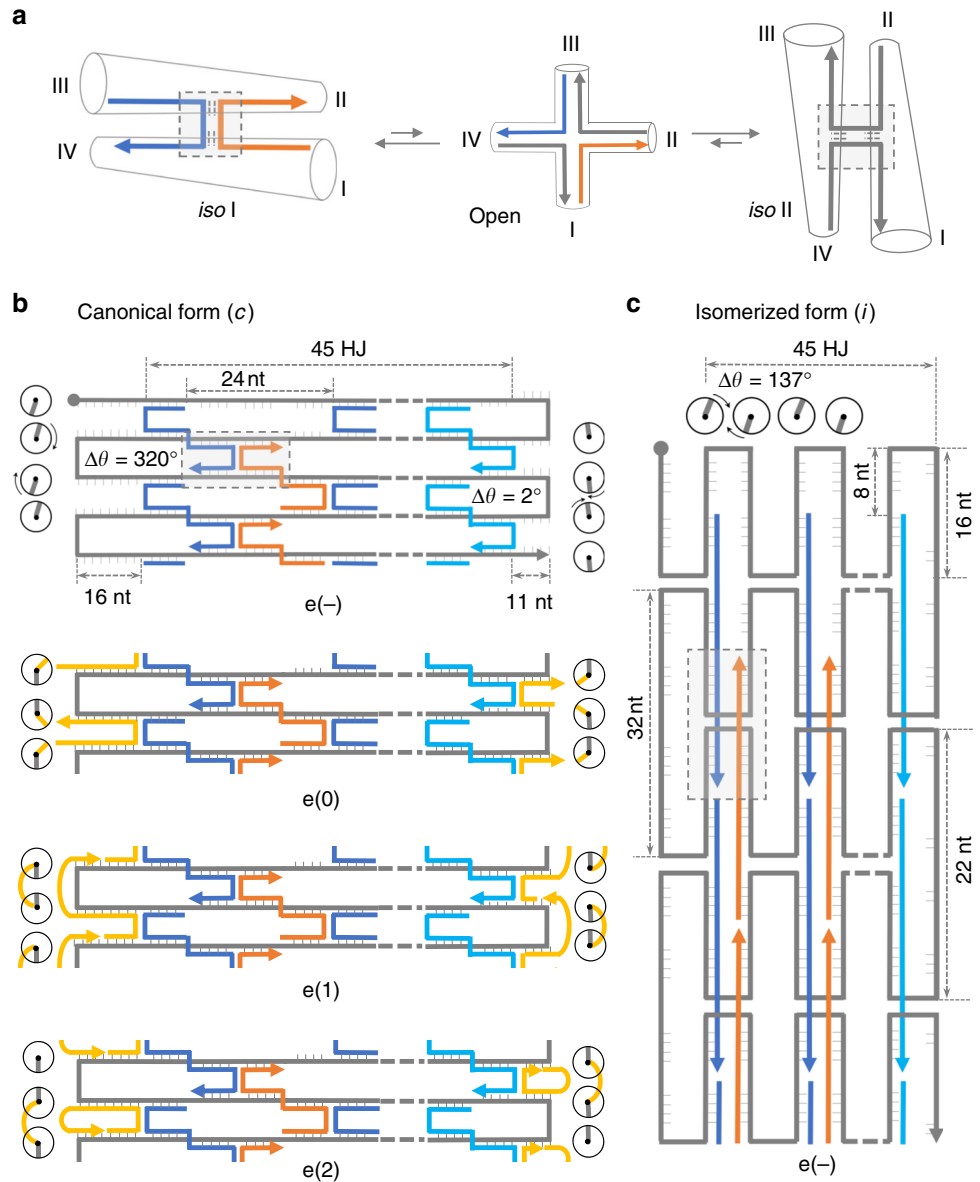

**Fig. 1** Design of three quasi-independent DNA origami domains. **a** In presence of magnesium ions, the Holliday junction (HJ) displays two isoforms, characterized by pairwise coaxial stacking of arms into a right-handed antiparallel stacked X-structure, with arms I/IV and III/II stacked in *iso* I and II/I, IV/III stacked in *iso* II. The interconversion between the two isomers passes through a less stable open intermediate, where the four helical arms point towards the corners of a square. Global *iso* I (**b**) and *iso* II (**c**) conformation of the HJs constituting a DNA origami domain and corresponding to the canonical (*c*) and isomerized (*i*) shape of the structure. In the *iso* I isomer the lateral loops are 32 nt- and 22 nt-long, respectively, at the left- and right-side of each domain, whereas 24 nt-long stretches of the unpaired scaffold are located at the top and bottom of each domain (e(−) construct). In the *iso* II form, instead, every domain adopts an "inverted" configuration, in which the HJs of the core (blue and orange strands) are flanked by lateral stretches and vertical loops of the unpaired scaffold (**c**). Note the last row of staples on the right-side of each domain (cyan strands): Although forming 32 bp-long duplexes, their secondary structure is partially constrained into a loop by a parallel stretch of unpaired scaffold that is 10 nt shorter. Both the left- and right-side of each domain have been addressed with three types of edges (yellow strands, in the e(0), e(1), and e(2) constructs), differing in the mechanical strain applied at the points of scaffold inversion. Circular symbols indicate the relative orientations of the bases at the helical edges (scaffold and staple bases are in gray and yellow, respectively). The angular displacement from the ideal situation in which the two scaffold bases are aligned and face each other (0°) denotes the degree of torsional stress generated by the hybridization of edge staples

O-shape, respectively (Suppl. Figs. 5 and 6). The three types of edges are therefore expected to apply different extents of mechanical strain at the scaffold turns.

We thus analyzed the effect of scaffold-turns position and mechanical strain in geometrically identical but chemically different scenarios, in various experimental conditions, for a total of about 50 constructs. The data revealed both topology- and sequence-dependent contributions to DNA origami assembly,

providing a deeper view of their interplay in the energetics and adaptability of the process.

**Topology-dependent contributions to DNA origami folding.** Our study mostly relies on the observation of the assembly products by atomic force microscopy (AFM). We noted that, in conditions of thermodynamic equilibrium, each domain may

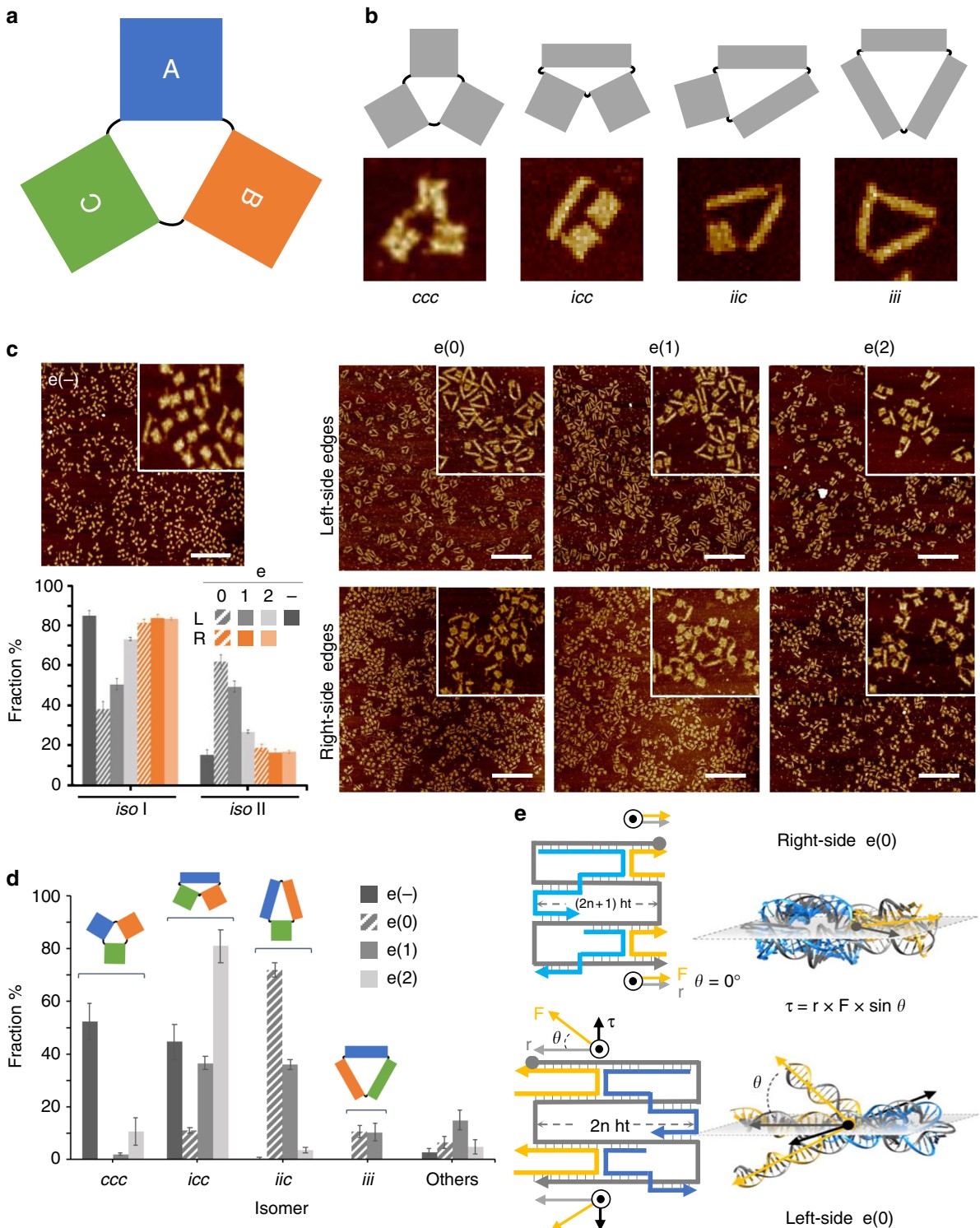

adopt one of two possible shapes: a canonical (*c*) or an isomerized (*i*) form, respectively associated to a global *iso* I or *iso* II state of the constituent HJs (Fig. 2b; Suppl. Figs. 7–9). The two isomeric shapes have approximately the same surface area but a different aspect ratio, with *iso* II being almost half wide and double as long than *iso* I (Suppl. Figure 2).

The entire origami structure can assume one of eight possible shapes, where each of the three domains can fold into one or the other isomer (Suppl. Fig. 10). These shapes are indicated here by a three-letter code starting from domain A and reading clockwise, such that an "*icc*" isomer will refer to a structure whose domain A

is isomerized (*iso* II), whereas domains B and C are present in the canonical (*iso* I) form. When the identity of the single domains cannot be uniquely assigned, the number of distinguishable species reduces to four (Fig. 2b) and the reading frame loses its significance, such that an isomer of the type "*icc*" will be formally equivalent to a "*cic*" or "*cci*" isomer.

We initially investigated the effect of scaffold-turns position on the assembly process and compared the results obtained in absence of edges (e(−); Fig. 2c left panel) with the structures obtained after addition of edges type 0, 1, or 2, either at the left- or right-side of all domains (Fig. 2c, e(0), e(1) and e(2), top and

**Fig. 2** Topology-dependent contributions to the folding of a DNA origami. **a** The full DNA origami structure is composed of three quasi-independent domains (A, B, and C) of identical shape but distinct sequence content, leading to four types of global isomers (**b**), wherein only the conformation, but not the identity, of the individual domains can be distinguished (c or i, for canonical or isomerized, respectively). **c** AFM characterization of the global structure in absence (e(−) construct) or presence of edges type 0, 1, or 2 (e(0), e(1), and e(2) constructs), either on the left- (L) or right-side (R) of each domain (upper and lower panels, respectively). Scale bars are 600 nm. Insets are 1 μm × 1 μm. Statistical AFM analysis of the end products is reported in the bar diagram as fraction of domains observed in the iso I or iso II shape (e(−) construct in dark gray bars; left- and right-side edges are indicated respectively in light gray and orange bars). **d** Statistical AFM analysis of the full DNA origami structure in the absence (e(−)) or presence of both the left- and right-side edges of type 0, 1, or 2. The highest degree of isomerization is given by design e(0) with ca. 11%, 72%, and 11% of global structures appearing as icc, iic, and iii isomers, respectively, and corresponding to ca. 62% of total iso II domains. **e** Atomic models of the e(0) edges on the right- and left-side of each domain (upper and lower panel, respectively) suggest that hybridization of edge staples to the scaffold generates a torque (**τ**) that is proportional to the length of the lever arm (gray vector, **r**), the force applied (yellow vector, **F**) and their angular displacement (θ) from the ideal coplanar position (for which θ = 0). **r** and **F** modules correspond to the lengths of the duplexes formed by the edge staple and the scaffold and located respectively upstream or downstream the crossover. Error bars were obtained from three independent images of the same sample, each showing several hundreds of individual structures. Source data are provided as a Source Data file

bottom panels, respectively). Close inspection of the end products by AFM clearly showed that in absence of edges about 15% of all domains appeared in the isomerized form (Fig. 2c, dark gray bars for e(−) in the diagram; Suppl. Fig. 11 and Suppl. Table 1). Addition of the right-side edges led to a minimal increase in the fraction of isomerized domains in all designs analyzed (ca. 17–19%, orange bars in the diagram of Fig. 2c). Contrarily, targeting the left-side scaffold turns led to a significant increase of isomerization, till about 62% in presence of the e(0) edges (striped gray bars in Fig. 2c and Suppl. Table 1). This confirms the hypothesis that scaffold inversions at unfavorable helical positions, such as those located on the left-side of each domain, become spots of high local frustration, whose bypassing drives the structure in search of an alternative folding pathway.

We also noted that the extent of isomerization resulting from addition of left-side triggers was extremely dependent on the type of edges formed, with e(0) and e(2) giving, respectively, the highest (62%) and lowest (29%) fraction of iso II domains (Fig. 2d and Suppl. Table 2). Thus, despite all triggers impose scaffold turns at identical positions, the degree of topological stress derived from their insertion in the structure must be different in the three designs.

A convenient way to describe the mechanical strain at the edges of a DNA origami structure is to indicate the orientations assumed by the terminal scaffold bases in absence of any constraint, i.e., assuming formation of a B-DNA duplex[27], and estimate the total angular displacement necessary to align them, enabling the scaffold to turn upon hybridization of edge staples (circular symbols in Fig. 1b and c; Suppl. Fig. 12). Accordingly, formation of e(0) in the iso I form requires a global rotation of 320° and 2° at the terminal-nucleobases located, respectively, on the left and right side of the structure, clearly indicating the higher torsional strain of the former over the latter turns (Fig. 1b, cfr. e(−) with e(0) constructs). The unsustainable mechanical stress on the left-side of the structure may be therefore released through structural reconfiguration of the edge and allosteric propagation to the entire domain, leading to the formation of the iso II isomer. This configuration is indeed accompanied by a total angular displacement of only 137° and is, therefore, more prone to tolerate the torsional strain applied (Fig. 1c). In line with this reasoning, the lower level of isomerization observed for e(1) and e(2) (respectively, 47% and 29%; Suppl. Table 2) can be ascribed to the presence of a flexible T5 loop at the point of scaffold turn, which may better accommodate different base orientations still preserving their connection (yellow curved strands in Fig. 1b).

Similar conclusions were inferred by molecular models[28] of two connected HJs emulating the topology of the edges in the different designs (Fig. 2e). The final view is that of a mechanical torque acting at the crossover joint, with the lever arm (**r**) and applied force (**F**) vectors corresponding to the duplex regions formed by the edge staple and the scaffold, positioned respectively upstream and downstream the crossover and pointing towards the boundaries of the structure (gray and yellow vectors in Fig. 2e). For permitted scaffold turns, as those on the right-side of the structure, the lever and force vectors of two vertically aligned HJs lie on the same plane and no global torque is applied at the crossovers (Fig. 2e, top panel). On the contrary, non-permitted scaffold inversions, as on the left-side of the structure, generate two out-of-plane force vectors oriented in opposite directions, i.e., above and below the plane of the HJs (Fig. 2e, bottom panel). This generates two torques (τ; black vectors in Fig. 2e), perpendicular to the lever and force arms and directed towards opposite directions on the plane of the HJs, causing the mechanical reconfiguration of the entire construct. In our models, the e(0) design displays a higher degree of torsional stress than e(1), in perfect match with the experimental results obtained (Suppl. Fig. 13). Correspondingly, a reduction in the length of the lever and force vectors, achieved through shorter edge staples, led to a dramatic decrease in the isomerization extent as a consequence of a weaker torque at the crossovers (from ca. 62% to roughly 19%; Suppl. Fig. 14 and Suppl. Table 3), thus confirming the validity of our mechanical model. The surprisingly low level of isomerization observed in the e(2) construct (ca. 29%) might be due to the almost prohibited circularization of the edge staples in the iso II form (Suppl. Fig. 9), thus promoting dimerization as an alternative way of structure folding (Suppl. Fig. 15).

Altogether, these experiments showed that the equilibrium position of a DNA origami assembly can be moved towards the iso II state by destabilizing the topology of the iso I state. We then tested whether the same result could be obtained by an opposite action, i.e., by stabilizing the topology of the iso II form. For this purpose, we reasoned that the removal of the last row of staples on the right-side of the structure (cyan strands in Fig. 1b and c), should reduce the degree of mechanical stress experienced by iso II, leading to a higher probability of its formation (Suppl. Fig. 16). Assembly of the origami domains lacking those staples fully confirmed our hypothesis, showing an increase of iso II shapes both in the absence and presence of e(0) triggers on the left-side (Suppl. Fig. 17 and Suppl. Table 4).

Addition of edges type 0 to a pre-assembled core structure resulted in a minimal increase in the percentage of isomerized domains, demonstrating that effective incorporation of these strands probably occurs at an early stage of the assembly process, even though they do not target contiguous regions of the scaffold that are statistically more easily accessible (Suppl. Fig. 18). Increasing the total DNA concentration and/or the annealing time did not show a large impact on the relative distribution of the isomeric species (Suppl. Figs. 19 and 20 and Suppl. Table 2),

supporting the view that DNA origami assembly is a thermo-dynamically driven process, whose outcome is strongly affected by the degree of structural frustration experienced in the initial assembly stage at the points of scaffold turn. Excessive strain at these topologically fragile sites may induce the global reconfiguration of all constituent HJs from an *iso* I to an *iso* II form. Nevertheless, the occurrence of isomerization in the absence of trigger strands (e(−) construct in Fig. 2c), that is, without application of mechanical stress, suggests that other factors, besides topology, may guide the folding pathway of DNA origami structures.

**Sequence-dependent contributions to DNA origami folding.** To unravel possible sequence-dependent contributions, we analyzed the end products obtained by folding each of the three domains individually, both in absence and presence of edges type 0 (Fig. 3a and Suppl. Table 5). Comparing the isomerization level of each domain (Fig. 3b) with the values obtained in the global structure (Fig. 2d), we noticed the formation of only four out of the eight possible isomers: namely, the fully canonical (*ccc*), the fully isomerized (*iii*) and two partially isomerized species, where only domain A (*icc*) or both A and B (*iic*) were present in the *iso* II form (Suppl. Fig. 10). This allowed to assign a conformational preference to every single domain, enabling to disclose the role of sequence content in patterns of HJs exposed to identical mechanical forces. In particular, in the absence of edges, domain A showed a 41% rate of isomerization, contrarily to B and C which were almost completely in the canonical form (fully colored bars in Fig. 3b and Suppl. Table 5). Addition of e(0) edges, moved the equilibrium nearly entirely towards the isomerized forms, both for A and B (94% and 85%, respectively indicated by blue and orange striped bars in Fig. 3b). Unexpectedly, domain C was hardly affected by the added mechanical stress and maintained its previous canonical form in more than 80% of the structures (green striped bars in Fig. 3b). These results indicate that origami domains made by an identical pattern of HJs with different base sequences may follow completely distinct folding trajectories, even if subject to the same mechanical forces in equal assembly conditions. In other terms, the high local frustration at the edges of the structure is necessary but not sufficient to trigger alternative folding pathways, meaning that the chemical information encoded in these critical regions plays the key decisional role that dictates whether a path will be traveled or not.

This hypothesis was confirmed by the following sets of data. First, we observed a consistent linear increase in the fraction of *icc* forms when assembling the full structure in presence of increasing $Mg^{2+}$ concentrations, from 2 mM to 20 mM (Fig. 3c; Suppl. Figs. 21–23 and Suppl. Table 6). Previous reports on the post-assembly reconfiguration of individual HJs claimed an effect of magnesium ions concentration on the kinetics of isomers' transformation but not on their equilibrium ratio[29]. This was shown to be instead mainly determined by the local DNA sequences at the junction[29,30]. Accordingly, our findings suggest that the folding process is biased towards the *iso* II state by a sequence-dependent factor, particularly evident in domain A and whose extent increases with higher $Mg^{2+}$ concentrations.

In a second experiment, we investigated the impact of individual edge strands on the assembly fate of domain A (Fig. 3d). For this purpose, the global e(−) design was assembled in presence of distinct pairs of e(0) staples, which targeted adjacent scaffold turns on the left-side of A (indicated from 1 to 9 in Fig. 3d and Suppl. Fig. 24). The addition of only one pair of edge staples was sufficient to observe an increase in the ratio of *icc/ccc* isomers, with a maximal conversion rate for the staples at positions 3/4 and a minimal effect for staples at positions 8/9 (respectively, 90% and 47% of *icc* isomers; Fig. 3d and Suppl.

Table 7). This suggests that identical mechanical forces acting on distinct chemical environments may have completely different consequences on the assembly pathway, reinforcing the view that topological stress alone cannot be responsible for the final fold of a DNA origami domain, but that the sequence content of the triggers and/or immediately adjacent strands plays an important and even more decisive role. An additional point emerging from these data is that the fraction of isomerized species obtained by addressing the entire left-side edge of domain A (e(0) A construct, striped blue bar in Fig. 3d) is not equal to the sum of the fractions observed by triggering the individual portions of the edge (gray bars in Fig. 3d), confirming a cooperative mechanism of conformational change mediated by the mechanical coupling of connected HJ motifs[15].

To understand this isomerization trend in detail, we estimated the energy of hybridization to the scaffold ($T_m$ values calculated by Mfold)[31] and the base stacking contributions[32,33] of each edge staple of domain A in the two possible isomeric states (Fig. 3e; Suppl. Figs. 25 and 26). The data obtained showed a good match between the melting temperatures of the staple pairs and the degree of isomerization caused by their presence in the assembly mixture (cfr. gray bars in Fig. 3d with colored bars in Fig. 3e). Contrarily, the calculated values of base stacking energies were not supportive, although suggesting a favorable *iso* II conformation in all cases (gray bars in Fig. 3e). The full picture can be therefore depicted as follows: Edge staples with higher $T_m$ (as the 1/2 or 3/4) hybridize to the scaffold in an early phase of the folding process and thus play a predominant role in the formation of the nucleation seeds from which the assembly starts and proceeds[21]. The topological strain experienced by these seeds is the key point of the assembly process, during which one of the two isomers may become energetically competitive, setting down the final isomerization ratio. On the other hand, staples of identical topology and mechanical action but lower thermal stability (such as the 5/6 or 8/9) will participate later in the process and mostly follow the ongoing pathway dictated by the leading strands.

We also analyzed the thermal GC maps[34] of every e(0) domain in each of the two isomeric forms (Suppl. Fig. 27) and constructed the corresponding base-stacking maps using the values of stacking energy previously reported in the literature[32,33] (Suppl. Figs. 28-30). The spatial distribution of GC pairs in the three domains well matched with the conformational responsiveness observed upon addition of edge staples, in full agreement with our sequence-dependent hypothesis of origami folding. Again, analysis of base stacking maps did not provide any consistent correlation to our experimental data, suggesting either a minor role of base stacking energies in origami folding or an inaccurate assessment of their value (Suppl. Fig. 31 and Suppl. Table 8).

**Tracing the energy landscape of DNA origami folding.** Our next goal was to quantify the energy cost for folding each of the two isomers, in order to better understand the reasons for the prevalence of one form over the other in the various scenarios explored. For this purpose, we applied ensemble temperature-dependent FRET spectroscopy to monitor the assembly and disassembly of domain A in absence and presence of different edge staples. We labeled the core of domain A using two ortho-gonal FRET systems: one specific to *iso* I and the other specific to *iso* II, thus enabling to differentiate the thermal profile of one isomer from the other (Suppl. Figs. 32–36 and Suppl. Table 9). The results are shown in Fig. 4. The canonical *iso* I shape of domain A appeared either in absence of edges or with edges type 2 (e(−) and e(2) profiles of Fig. 4a), while the *iso* II form occurred in all designs analyzed (Fig. 4b), in full agreement with our

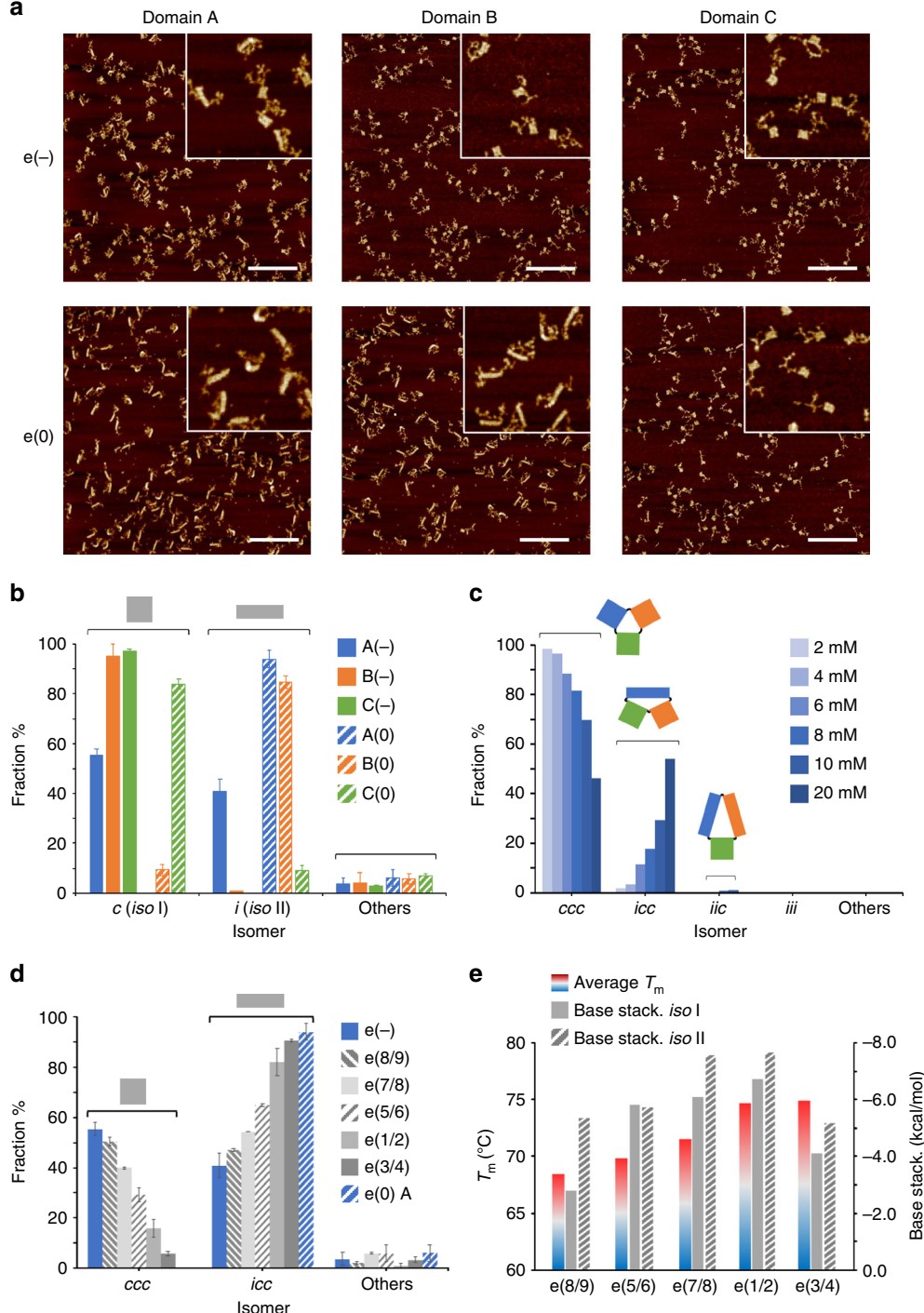

**Fig. 3** Sequence-dependent contributions to the folding of a DNA origami. **a** Representative AFM images and **b** statistical analysis of the end products obtained by assembling each individual domain either in the absence or presence of edge staples of type 0 (e(−) and e(0) constructs, respectively indicated by full and striped colored bars in blue, orange and green, for domain A, B, and C). Scale bars of the AFM images are 600 nm. Insets are 1 μm × 1 μm. **c** Effect of $Mg^{2+}$ ions concentration on the distribution of the end products, indicating a bias for the *iso* II form of domain A enhanced by increasing magnesium concentrations. **d** Statistical analysis of the end products obtained by assembling domain A in the e(−) design in presence of distinct pairs of e(0) staples (indicated as 1/2, 3/4, 5/6, 7/8 and 8/9), targeting consecutive scaffold turns on the left-side of the structure. **e** Thermal stabilities (colored bars) and base stacking energies of the e(0) staple pairs of domain A in both the *iso* I and *iso* II conformation (full and striped gray bars). Note the good correlation between the average $T_m$ of the staples and the degree of isomerization induced (d). A poor match results instead by the calculated base stacking contributions. Error bars were obtained from three independent images of the same sample, each showing several hundreds of individual structures. Source data are provided as a Source Data file

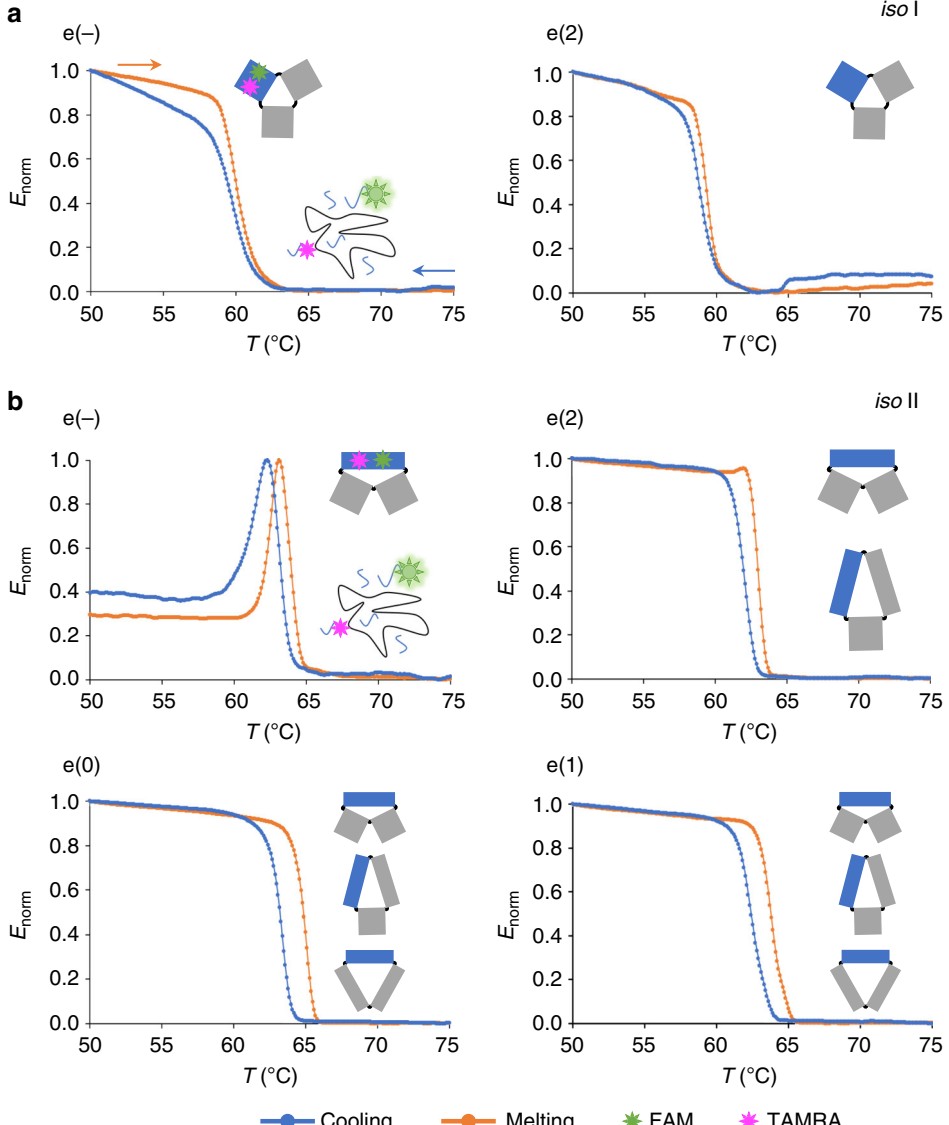

**Fig. 4** Thermal-dependent ensemble FRET studies. Normalized FRET efficiency profiles for the assembly and disassembly of domain A, either in absence of edges (e(−) construct) or in presence of edges of type 0, 1, or 2 (e(0), e(1) and e(2) constructs). Two FRET labeling strategies (FRET I and FRET II) have been used to monitor exclusively the assembly/disassembly of the *iso* I (**a**) or *iso* II (**b**) conformer, using distinct arrangements of FAM- and TAMRA-labeled oligonucleotides belonging to the core of domain A (green and magenta symbols, respectively). All samples were exposed to cooling (blue curves) and melting (orange curves) in the range between 25 °C and 85 °C with a rate of ±0.1 °C/min. Data were averaged from four replicates of two independent experiments. The possible global isomers resulting from every edge design are schematically represented (domain A is indicated in blue). Note the quasi-reversibility of the *iso* I conformer and the hysteresis of the *iso* II form (shifted to higher $T_m$), particularly the biphasic profile of this latter in the e(−) construct. Source data are provided as a Source Data file

structure assignment on the basis of AFM imaging (Fig. 2d). Formation of the *iso* I structure was nearly cooperative and reversible with a $T_m$ of about 60 °C (Table 1 and Suppl. Figs. 37-39). Contrarily, the *iso* II state was characterized by ca. 1–2 °C of hysteresis centered at a higher $T_m$ (around 63 °C to 64 °C). This clearly indicates that whereas the assembly of *iso* I is under thermodynamic control, formation of *iso* II is not a reversible process and is typically associated to a higher energetic cost. Particularly interesting is the thermal behavior of the e(−) construct in the *iso* II form (Fig. 4b, e(−) design). Besides hysteresis, this structure is characterized by a biphasic profile: full assembly of the *iso* II at ca. 63 °C (associated to a 1.0 FRET signal) is followed by a rapid loss in energy transfer till a value of ca. 0.4 at the same temperature observed for *iso* I formation (Fig. 4a). This suggests that, in absence of mechanical strain at the edges, the *iso*

II state is favored in the initial phase of the assembly process; however, once formed, it undergoes a structural reconfiguration to *iso* I in almost 60% of the structures, in full agreement with the results observed by AFM imaging (full blue bars in Fig. 3b). Similar conclusions were drawn from thermal experiments performed at a different annealing rate, again confirming that the two isomers are not the result of a kinetically trapped process, rather the endpoints of two distinct but connected folding pathways (Suppl. Figs. 40–42 and Suppl. Table 10). Differently from *iso* I, the lack of reversibility in the *iso* II forms prevented the application of the van't Hoff equation to extract the thermodynamic parameters of the process. We, therefore, used the Friedman-Ozawa isoconversional method8[35-38] to estimate the activation energies of the assembly and disassembly events in a model-independent fashion (Suppl. Note 1 and Table 1). The e(2)

**Table 1 Energetics of assembly/disassembly of *iso* I and *iso* II forms**

| | *iso* I | | | | | *iso* II | | | | |
|---|---|---|---|---|---|---|---|---|---|---|
| | $T_m^{U\to I}$ (°C) | $E_{act}^{U\to I}$ (kcal/mol) | $T_m^{I\to U}$ (°C) | $E_{act}^{I\to U}$ (kcal/mol) | $\Delta E_{\to I}$ (kcal/mol) | $T_m^{U\to II}$ (°C) | $E_{act}^{U\to II}$ (kcal/mol) | $T_m^{II\to U}$ (°C) | $E_{act}^{II\to U}$ (kcal/mol) | $\Delta E_{U\to II}$ (kcal/mol) |
| e(−) | 59.9 | 250 ± 69 | 59.8 | 348 ± 55 | −98 | 63.2; 61.8 | - | 63.7; 62.4 | - | - |
| e(0) | - | - | - | - | - | 63.5 | 488 ± 174 | 65.0 | 432 ± 150 | 56 |
| e(1) | - | - | - | - | - | 62.3 | 378 ± 83 | 63.7 | 469 ± 77 | −91 |
| e(2) | 58.8 | 375 ± 77 | 59.3 | 452 ± 67 | −77 | 62.1 | 492 ± 124 | 63.0 | 971 ± 169 | −479 |

Thermodynamic and kinetic parameters of the thermal assembly (U→I and U→II) and disassembly (I→U and II→U) of domain A derived by analysis of the temperature-dependent FRET profiles in both the FRET I and FRET II labeling strategies. The melting temperatures ($T_m$, in °C) have been calculated from the first derivative of the thermal profiles. The $T_m$ values for the cooling and melting process of individual constructs are approximately equal in the *iso* I form, while they differ of about 2 °C in the *iso* II form, denoting the higher reversibility of the former over the latter species. Activation energies ($E_{act}$, in kcal/mol) from the unfolded (U) to the folded state (I or II) and *viceversa* have been calculated applying the model-free isoconversional Friedman-Ozawa method (see Suppl. Note 1). Mean energies and standard deviations were obtained from sets of 50 to 450 values, whose linear correlation was larger than 0.98. Not accessible data are indicated by a (-) symbol.

design gave treatable thermal profiles in both conformational states, enabling to trace the full energy landscape of the thermal process and offering a paradigm of origami assembly with two possible outcomes (Fig. 5a). Starting from a mixture of staples randomly distributed in solution (unfolded state, U), the lower energy barrier to the *iso* I state makes the assembly of this species kinetically favored over the *iso* II form ($E_{act}^{U\to I} < E_{act}^{U\to II}$; Fig. 5b and Table 1). The reverse process, i.e., unfolding, is also more probable for the *iso* I state, due to the lower energy barrier that must be overcome for disrupting the folded structure ($E_{act}^{I\to U} \ll E_{act}^{II\to U}$; Table 1). The entire landscape thus presents two minima of energy, one associated to the *iso* I and the other to the *iso* II state. The former is more easily accessible and prone to reversibility, while the latter is typically less probable and characterized by a larger hysteresis: that is, when formed, it reaches a minimum of energy from which it cannot easily escape (Fig. 5b). Finally, as previously reported for single HJs and DNA origami structures[18,29], the two isomers are related by a post-assembly route that describes the structural reconfiguration from one to the other state passing through an intermediate open form. This post-assembly transformation has been further confirmed by single-molecule force spectroscopy measurements with optical tweezers, performed on domain A in the e(−) design (Suppl. Figs. 43–44). The constant trap-distance experiments at a pre-tension of around 5 pN revealed the existence of three interconverting states, which we identified as the *iso* I, *iso* II and the open form, with this latter being more easily accessible from the *iso* I rather than the *iso* II shape, in agreement with our general folding model (inset of Fig. 5b).

## Discussion

The self-assembly of a DNA origami structure, even of small size and simple shape, relies on the perfect matching of thousands of weak non-covalent interactions between a few hundreds of distinct single strands and a unique scaffold sequence. Despite the challenging construction principle, the robustness of the method undoubtedly demonstrates that an efficient folding pathway, although not evolutionary tailored, must indeed exist. Recent studies have evidenced the important role of the scaffold turns as guiding sites of initial structural shaping[14,39] and topological targets of post-assembly reconfiguration[15,17,18]. On another front, high-temperature nucleation sites have been also identified as initiators of thermal folding[21], although not necessarily located along contiguous regions of the scaffold.

Our work merges topological and sequence-dependent views into a unique scenario and provides quantitative data of activation energies to trace the full energy landscape of small DNA origami domains. In our model, the assembly fate is dictated by the connectivity of the nucleation seeds at the very beginning of

the folding process, that is, by the degree of torsional strain experienced at the initiation sites during their formation and growth. This feature is captured in the energy barrier to the *iso* I or *iso* II folded state and may be tentatively explained as a compromise between an enthalpic and an entropic contribution. Whereas the former relies on hydrogen bonds formation, base stacking interactions and elastic deformation of DNA during the assembly process, the latter mostly depends on the shape-related constraints of the scaffold in the growing object. Thus, although the higher number of base stacking interactions may play in favor of the *iso* II state, the large elastic and entropic costs associated to its formation strongly reduce its probability of folding with respect to the structurally more relaxed *iso* I isomer (Fig. 5b). The situation however completely changes when the folding process initiates at highly frustrated scaffold turns. In this case, the energy barrier to the *iso* I state may become prohibitively high, enabling the competitive formation and progressive accumulation of the otherwise unfavored *iso* II structure. The formation of topologically stressed nucleation sites thus reshapes the energy landscape of the assembly process, affecting the relative populations of the two isomers to an extent that is proportional to the degree of mechanical stress experienced.

In conclusion, we here propose a dynamic model of DNA origami assembly that adapts in a complex way to the thermal stability and structural connectivity of the staples from which the nucleation process initiates, recapitulating both the folding path to one of two distinct isomers as well as their post-assembly reconfiguration. However, differently from proteins, the self-recognition properties of the DNA, together with the entropic advantage of a scaffold-driven assembly, tremendously accelerate the responsiveness of the system to external stress, enabling to bypass high energy barriers and explore alternative folding pathways.

## Methods

**Materials and chemicals**. All unmodified oligonucleotides were purchased from Sigma-Aldrich as desalted products and delivered lyophilized in 96-well plates. 6-carboxyfluorescein (FAM) and carboxytetramethylrhodamine (TAMRA) oligonucleotides were purchased from Sigma Aldrich in HPLC purification grade and used without further treatment. Single-stranded M13mp18 DNA, was produced from phage DNA (Affymetrix) and propagated in *E.coli* XL1-Blue (Agilent technologies). Buffers used were 1× TEMg (20 mM Tris base, 2 mM EDTA, 12.5 mM MgCl₂, pH 7.6), 1× TBEMg (40 mM Tris base, 20 mM boric acid, 2 mM EDTA, 12.5 mM Mg acetate, pH 8.0) and 1X TBE (89 mM Tris base, 89 mM boric acid, 2 mM EDTA, pH 8.0).

**DNA design and sample preparation**. DNA origami structures were designed with caDNAno (www.cadnano.org) and assembled using a 1:50 molar ratio between the M13mp18 ssDNA scaffold (2 nM) and each of the staple strands, in 1× TEMg buffer. Thermal annealing was performed by decreasing the temperature from 80 °C to 20 °C at −1 °C/min on a Thermocycler Mastercycler nexus gradient (Eppendorf), upon an initial denaturation at 80 °C for 5 min. The reaction mixtures

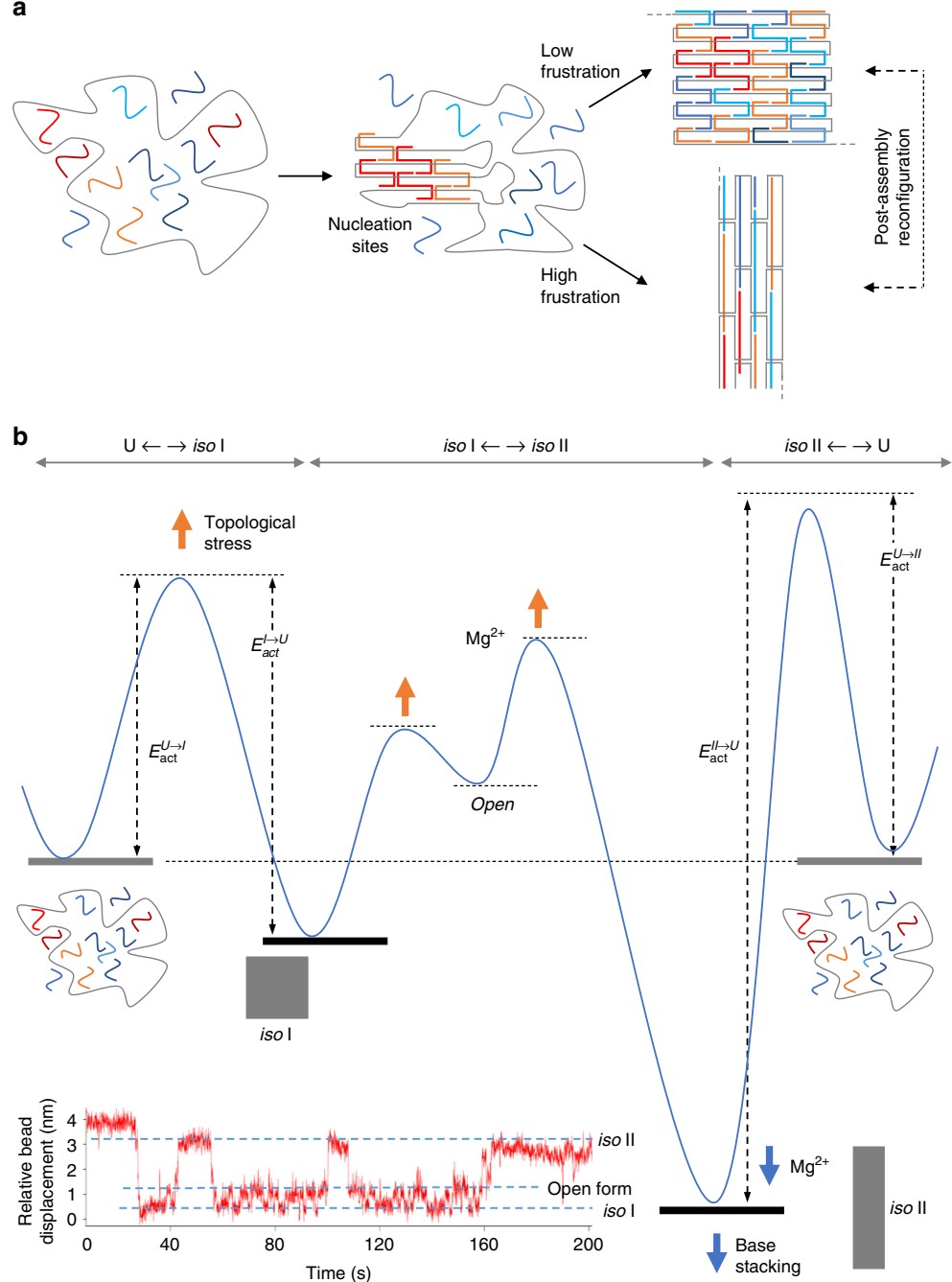

**Fig. 5** Energy landscape of DNA origami assembly. **a** In our proposed dynamic model of DNA origami folding, the final fate is dictated by the degree of topological stress experienced at the nucleation sites. Besides the formation of the *iso* I state, favored by a low level of structural frustration, this mechanism enables to travel an alternative folding pathway leading to a typically less populated *iso* II state, which becomes competitive for structurally frustrated nucleation sites. **b** Assembly pathway of domain A in the e(2) design according to the values of activation energy for the cooling and melting of both the *iso* I and *iso* II isomers. The energy landscape is representative of a DNA origami assembly with two possible outcomes, connected by a post-assembly reconfiguration that passes through an intermediate open form, as shown by single-molecule force spectroscopy measurements (inset). The energy barriers for folding/unfolding of the *iso* I species are lower than those of the *iso* II, resulting in the easier formation and reversibility of the former and large hysteresis of the latter. The lower rate of *iso* II formation might be due to the entropic penalty and elastic energy cost associated to its folding. Here, the scaffold is entangled into a highly ordered structure constituted by many duplexes shorter than the persistence length of double-stranded DNA. Nevertheless, once the kinetic barrier is passed, the energy gain appears to be large enough to reach an absolute minimum from which the structure cannot easily escape. The energy level associated to the open form has been set arbitrarly and is not matter of this study. Note the two-fold effect of magnesium ions on the energy landscape of connected HJs: increasing $Mg^{2+}$ concentrations lead to a decrease in the *iso* I/*iso* II conversion rate (orange arrows, previous studies[29]) and stabilization of the *iso* II form (blue arrows, this study)

were then analyzed by agarose gel electrophoresis using 0.75% agarose (Biozym) in 1× TBEMg, at 80 V for 2 h at 4 °C. 1 kbp DNA ladder was purchased from Roth. The gel was scanned with a Typhoon FLA 9000 (GE healthcare Life Sciences) at different wavelengths and finally stained with ethidium bromide (Merck). The full list of sequences is provided in Supplementary Data 1.

**AFM imaging.** The sample (5 µl)was deposited on freshly cleaved mica surface (Plano GmbH) and adsorbed for 3 min at room temperature. After washing with ddH$_2$O, the sample was dried under gentle argon flow and scanned in ScanAsyst Mode using a MultiMode$^{TM}$ microscope (Bruker) equipped with a Nanoscope V controller, using cantilevers with sharpened pyramidal tips (ScanAsyst-Air tips, Bruker). Biotinylated samples were further treated with a large molar excess of streptavidin. Several AFM images were acquired from different locations of the mica surface to ensure reproducibility of the results. All images were analyzed by using the NanoScope Analysis 1.5 software.

**Temperature-dependent FRET spectroscopy.** Thermal experiments were performed using a CFX96 real time system (BioRad), using excitation and emission wavelengths for fluorescein detection. DNA origami samples were prepared from 50 nM M13mp18 viral DNA, 250 nM staple strands and 100 nM fluorescent modified sequences in 1× TEMg buffer. The assembly/disassembly of the DNA origami samples was monitored by consecutive thermal annealing/melting processes in the temperature range from 20 to 80 °C and a rate of ± 0.1 °C/min. Alternatively, a shorter assembly ramp, from 50 °C to 75 °C was used or a cooling rate of ± 1 °C/min. For each construct, both the donor–acceptor (FAM-TAMRA) and donor-only (FAM) samples were prepared, each in four copies and analyzed in two independent runs. Successful assembly of the structures was proven by atomic force microscopy.

**Kinetic analysis of the thermal processes.** The thermal curves were analyzed using a model-free method to determine the activation energy for the folding and unfolding process, from which a representative energy landscape has been traced (details of the analysis are given in Supplementary Note 1). Statistical treatment and analysis of the data has been performed using a custom-made software suite coded in R.

**Single-molecule force measurements.** Constant-trap-position experiments have been performed on a dual optical tweezer (C-Trap$^{TM}$ from Lumicks, The Netherland). Opposite helical domains of the DNA origami structure were linked to two double-stranded 3 kbp-long DNA handles, functionalized at the ends either with digoxygenin or biotin. Digoxygenin- and biotin-functionalized handles were obtained by PCR amplification of the pET-28a vector (5369 bp). DNA handles, bridging the DNA origami to the beads, contained a central PEG9 spacer flanked by two ca. 30 nt-long stretches, one partially complementary to the M13 scaffold and the other partially complementary to the tether. For the tweezers experiment, DNA origami-handles constructs were tethered between 1.76 µm streptavidin polystyrene beads (Spherotech, Inc) and 1 µm polystyrene beads (Spherotech, Inc), previously covalently functionalized with anti-digoxigenin Fab fragments (Roche). Measurements were carried out at room temperature in 1× TEMg buffer with the addition of an oxygen scavenger system (26 U/ml glucose oxidase, 17,000 U/ml catalase).

## Code availability
The programming code used to analyze the raw data that support the findings of this study is available from the corresponding author upon request.

## Data availability
The authors declare that the data supporting the findings of this study are available within the paper and its Supplementary Information files. Source data for Figures and Supplementary Figures are available in the Source Data file. Additional raw data are available from the corresponding author upon reasonable request.

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

## Acknowledgements

This work was supported by the Deutsche Forschungsgemeinschaft (DFG SA 1952/3-1) founded to B.S. We also thank Lena Winat for the development of the software for analysis of the data, Daniel Habermann, Lisa Mazul and Kira Stein for the experimental support and Jordi Cabanas-Danés for the kind assistance in single-molecule measurements and helpful discussions.

## Author contributions

R.K., W.P. and B.S. conceived the experiments; R.K. performed the DNA assembly, FRET thermal analysis, and AFM imaging and analyzed the data; A.M., A.C. and P.R. performed the single-molecule force experiments and analyzed the data; B.S. wrote the manuscript; all authors discussed the results and commented on the manuscript.

## Additional information

**Competing interests:** The authors declare no competing interests.

