## [Peer Review File · Nature Communications]

Reviewers' Comments:

Reviewer #1:

Remarks to the Author:

This exciting study merges the topological and sequence dependent views to study the folding of monolayer DNA origami and provides quantitative data to trace the full energy landscape of folding. To understand the topological constraints in folding of DNA origami, the authors designed reconfigurable DNA origami structures that has two energy minima states; canonical (iso I) state where antiparallel stretches of scaffold strand are kept in place by staple crossovers and iso II isomer state where antiparallel stretches of staples are joined by scaffold crossovers. The authors studied folding in the presence and absence of different types of edges staples to induce scaffold inversions adding topologically stressed sites. Additionally, to understand the role of sequence depending in folding, the authors studied folding of three quasi-independent domains of identical shape but distinct sequence content. The authors propose a dynamic model of DNA origami folding based on their observations that the formation of topologically stressed sites reshapes the energy landscape of the assembly process and dictates the populations of the two isomers. The authors identify the degree of mechanical stresses and their role in early stages of nucleation as key factors influencing the outcome of DNA origami folding.

The general subject of the study fits the journal scope quite well and will be useful to the broad scientific community. I recommend this article to be published in Nature Communications, after addressing the following minor comments:

1. Can the authors comment on the mechanical stresses/folding trajectories involved when folding a circular scaffold as compared to that of a linear scaffold?
2. Page 11, line 12: This latter was shown -> This later was shown....
3. Figure 2 caption. (d) -> (e)

Reviewer #2:

Remarks to the Author:

By bringing the possibility to program the assembly of DNA brick into virtually any kind of desired functional nanostructures, the so-called « DNA origami » is one of the most important self-assembly methods invented in the 21st century, with a particularly mighty impact in nanoscience. Because this method is exceptionally robust, high-yield and practically flawless, it has been adopted by a huge number of scientists. Surprisingly, very little is known about the folding pathways of DNA origamis, how to guide it and how to re-orientate it so that non-canonical shapes could be obtained. This is precisely these three challenges that are addressed by this impressive manuscript. By systematic and smart changes of edge staple compositions, the author reliably establish the probability reach different origami isomers as a function of the mechanical stress experienced during the initial folding steps. Very interestingly, they can further build an energy landscape of DNA origami, which is shown to be adaptive and eventually completely reorientate the folding toward structures that are otherwise unfavored. This not only dramatically increase our understanding of DNA origami folding process but provides also keys to redirect the folding into non-canonical shapes. The experimental design is particularly smart. The conclusions are very solid and based on a huge number of reliable data. The data analysis is also particularly complete. Overall, although very dense, the manuscript is well written and pleasant to read. For all of these reasons, I recommend the publication of this manuscript in Nature Communications. I just have a few suggestions of minor modification, and they all concern the presentation.

- 1) Although I find the manuscript rich and clear (introduction, results, conclusions, figures), I feel that the abstract does not convey well the manuscript content, by being a bit too general and too specific at the same time. It uses some technical terms (iso I and iso II isomers), which might be difficult to understand for a non-expert readers. I would also suggest to explain better the experimental approach through the use of edge staples to direct the folding.

2) In figures 2 and 3, I would suggest to plot the distribution of isomers obtained for each edge composition rather than the opposite, which for me would be more straightforward to analyze. This is of course just a question of presentation and I let the authors decide what presentation they estimate to be the most relevant.

3) Page 2, 23rd line, "to distinct" should be replaced with "two distinct"

4) Legend of Figure 2, 12th line: (d) should be replaced with (e)

Sites of high local frustration in DNA origami

Richard Kosinski, Ann Mukhortava, Wolfgang Pfeifer, Andrea Candelli, Philipp Rauch, and Barbara Saccà

We are extremely glad for the enthusiastic comments received from the Reviewers and the positive support of the Editor. We feel that our work has been sincerely appreciated and, with it, we hope to have contributed to advance our understanding of DNA origami self-assembly: a robust and widely-used method, still holding intriguing aspects.

Below are our point-by-point responses to the Reviewers' comments.

REVIEWERS' COMMENTS:

Reviewer #1 (Remarks to the Author):

This exciting study merges the topological and sequence dependent views to study the folding of monolayer DNA origami and provides quantitative data to trace the full energy landscape of folding. To understand the topological constraints in folding of DNA origami, the authors designed reconfigurable DNA origami structures that has two energy minima states; canonical (iso I) state where antiparallel stretches of scaffold strand are kept in place by staple crossovers and iso II isomer state where antiparallel stretches of staples are joined by scaffold crossovers. The authors studied folding in the presence and absence of different types of edges staples to induce scaffold inversions adding topologically stressed sites. Additionally, to understand the role of sequence depending in folding, the authors studied folding of three quasi-independent domains of identical shape but distinct sequence content. The authors propose a dynamic model of DNA origami folding based on their observations that the formation of topologically stressed sites reshapes the energy landscape of the assembly process and dictates the populations of the two isomers. The authors identify the degree of mechanical stresses and their role in early stages of nucleation as key factors influencing the outcome of DNA origami folding.

The general subject of the study fits the journal scope quite well and will be useful to the broad scientific community. I recommend this article to be published in Nature Communications, after addressing the following minor comments:

We sincerely thank this reviewer for the positive evaluation of our work.

1. Can the authors comment on the mechanical stresses/folding trajectories involved when folding a circular scaffold as compared to that of a linear scaffold?

We thank this reviewer for raising up this very good point. At the beginning of our study, we have indeed thought that the circularity of the scaffold could have been one source of unsustainable topological stress leading to formation of an alternative and mechanically more stable form. Accordingly, we reasoned that a linear scaffold should permit to release the excessive tension and enable formation of a higher amount of canonical structures. To prove this hypothesis, we introduced a nick into the circular scaffold at one selected site located within the core of domain A and performed the self-assembly of the full structure using the so-obtained linear scaffold and edge staples type 0. Our results are presented in the new Supplementary Figure 46. A description of the procedure used for the enzymatic restriction

and purification of the linear scaffold is reported in the Supplementary Materials and Methods section. In brief, we did not observe a significant increase in the fraction of canonical species, disproving our initial hypothesis, i.e. meaning that the circularization of the scaffold is not the main reason for the observed conformational change. In view of our further studies, we deduce that this is due to introduction of the nick into a region that is distant from the nucleation strands at the edges and has a lower thermal stability, thus affecting the fate of the assembly only in minimal part. The situation might be different when the nick is introduced at one of the scaffold turns, particularly if involved in the initiation of the assembly process. In this case, we presume that the conditions are met to release the mechanical stress applied without the need for isomerization of the entire domain.

2. Page 11, line 12: This latter was shown -> This later was shown....
Corrected into "This was shown to be ...".

3. Figure 2 caption. (d) -> (e)
Done.

Reviewer #2 (Remarks to the Author):

By bringing the possibility to program the assembly of DNA brick into virtually any kind of desired functional nanostructures, the so-called « DNA origami » is one of the most important self-assembly methods invented in the 21st century, with a particularly mighty impact in nanoscience. Because this method is exceptionally robust, high-yield and practically flawless, it has been adopted by a huge number of scientists. Surprisingly, very little is known about the folding pathways of DNA origamis, how to guide it and how to re-orientate it so that non-canonical shapes could be obtained. This is precisely these three challenges that are addressed by this impressive manuscript. By systematic and smart changes of edge staple compositions, the authors reliably establish the probability to reach different origami isomers as a function of the mechanical stress experienced during the initial folding steps. Very interestingly, they can further build an energy landscape of DNA origami, which is shown to be adaptive and eventually completely reorientate the folding toward structures that are otherwise unfavored. This not only dramatically increases our understanding of DNA origami folding process but provides also keys to redirect the folding into non-canonical shapes. The experimental design is particularly smart. The conclusions are very solid and based on a huge number of reliable data. The data analysis is also particularly complete. Overall, although very dense, the manuscript is well written and pleasant to read. For all of these reasons, I recommend the publication of this manuscript in Nature Communications. I just have a few suggestions of minor modification, and they all concern the presentation.

We are very grateful to this reviewer for the sincere appreciation of our work.

1) Although I find the manuscript rich and clear (introduction, results, conclusions, figures), I feel that the abstract does not convey well the manuscript content, by being a bit too general and too specific at the same time. It uses some technical terms (iso I and iso II isomers), which might be difficult to understand for non-expert readers. I would also suggest to explain better the experimental approach through the use of edge staples to direct the folding.

We agree with this referee and accordingly modified the abstract to make it more accessible to non-experts of the field.

2) In figures 2 and 3, I would suggest to plot the distribution of isomers obtained for each edge composition rather than the opposite, which for me would be more straightforward to analyze. This is of course just a question of presentation and I let the authors decide what presentation they estimate to be the most relevant.

We thank this reviewer for this comment and agree that both distributions, although being scientifically valid, may give different insights. Our data are always presented as a distribution of two (*iso I* and *iso II*) or four (*ccc*, *icc*, *iic* and *iii*) isomeric forms as a function of different design parameters, such as the position of the scaffold turns (left- *versus* right-side), the topology of the edge strands (type 0, 1, 2), their number and/or base-composition (domain A, B or C), as well as variable experimental conditions, such as magnesium ions concentration, total DNA content and annealing rate. Thus, during our systematic analysis, some experiments were performed keeping the edge type fixed and analyzing the distribution of shapes as a function of another parameter (e.g. Mg ions concentration or base-sequence). To keep homogeneity in the presentation of the data and enable easier comparison of the results, we decided to present our distributions always in the same way, with isomeric forms along the x-axis and populations fraction along the y-axis. However, we agree that this is only a matter of taste.

3) Page 2, 23rd line, “to distinct” should be replaced with “two distinct”

Thanks for pointing out this detail. We have changed to “to hybridize distinct and”, as it may also hybridize to more than two scaffold segments.

4) Legend of Figure 2, 12th line: (d) should be replaced with (e)

Done.